# Prevalence and Covariates of Electronic Cigarette Use among Students Aged 13–15 Years in the Philippines: 2019 Global Youth Tobacco Survey

**DOI:** 10.3390/ijerph20247193

**Published:** 2023-12-18

**Authors:** Chelsea Serra, Gibril Njie, Nerline Jacques, Liping Pan

**Affiliations:** 1Global Tobacco Control Branch, Office on Smoking and Health, National Center for Chronic Disease Prevention and Health Promotion, Centers for Disease Control and Prevention (CDC), Atlanta, GA 30341, USA; vow9@cdc.gov (G.N.); top4@cdc.gov (N.J.); lmp6@cdc.gov (L.P.); 2CDC Global Health Fellowship Program, Public Health Institute, Oakland, CA 94607, USA; 3CyberData Technologies, Herndon, VA 20171, USA

**Keywords:** e-cigarettes, Philippines, students, tobacco, youth, prevalence

## Abstract

Electronic cigarette use is growing in popularity and accessibility among youth in the Southeast Asia region. We analyzed data on 6670 students, aged 13–15 years, from the Philippines’ 2019 Global Youth Tobacco Survey. Prevalence estimates and 95% confidence intervals (CI) were estimated for current use (i.e., past 30 days), ever use, and awareness of e-cigarettes. Chi-square tests compared prevalence differences between groups. Multiple logistic regression models assessed factors associated with e-cigarette use while controlling for sociodemographic characteristics, current use of other tobacco products, and secondhand smoke exposure. Prevalence of current e-cigarette use was 14.1% (95% CI = 12.4%, 15.8%), ever use was 24.6% (95% CI = 22.4%, 26.9%), and awareness was 75.5% (95% CI = 73.0%, 78.0%). Current use of any other tobacco products and exposure to secondhand smoke at home, school, or other public places were positively associated with current and ever use of e-cigarettes. Boys and youth living in Luzon or Mindanao had higher odds of current e-cigarette use compared to girls and youth in Visayas. Findings indicated that one in four Philippine students aged 13–15 years ever used e-cigarettes and one in seven currently use e-cigarettes. This study highlights the importance of implementing evidence-based strategies, including relevant tobacco control policies.

## 1. Introduction

Tobacco kills up to half of the people that use it [1,2]. Globally, more than 8 million deaths result directly from tobacco use, while approximately 1.3 million deaths occur from exposure to secondhand smoke [2]. Electronic cigarette (e-cigarette) use is growing in popularity and accessibility among youth and adults globally [2,3]. On average, in 2012, global health care expenditure due to a smoking-related illness was USD 422 billion [3]. The Southeast Asia market, including countries such as Indonesia, Malaysia, Vietnam, and the Philippines, are expected to have significant growth in the e-cigarette market in coming years, making them a desirable target of tobacco companies [1]. According to a comprehensive literature review of current e-cigarette use among adolescents in Southeast Asia, the prevalence of current use ranges from 3.3% to 11.8% [4]. Tobacco use is the leading risk factor for mortality in the Philippines [5]. Studies show that students are more susceptible to e-cigarette use because of their flavors, attractive packaging, price promotions, and use among peers [6,7]. E-cigarettes are often marketed as a cessation device or substitute for cigarettes; however, they have the potential to increase nicotine addiction among young adults and youth who do not smoke [6,8].

Toxicants and carcinogens in combustible cigarette smoke remain higher than aerosols produced by e-cigarettes [8]. In 2005, the Philippines became party to the World Health Organization (WHO) Framework Convention on Tobacco Control (FCTC) to control tobacco use and exposure to tobacco smoke in the country [5,9,10]. To reduce tobacco demand, the WHO introduced the MPOWER measures (M: Monitor tobacco use and prevention policies, P: Protect people from tobacco smoke, O: Offer help to quit smoking, W: Warn about the dangers of tobacco, E: Enforce bans on tobacco advertising, promotion and sponsorship, R: Raise taxes on tobacco) [11]. The Philippines implemented regulations focused on the marketing and use of tobacco products [9,10]. However, as of 2019, of students that currently smoke cigarettes in the Philippines, 37.1% were not restricted when purchasing tobacco products and e-cigarettes [12]. A previous study based on pooled analyses of 68 countries, including the Philippines, found that the prevalence of current e-cigarette use was 9.2% among youth aged 12 to 16 years from 2012 to 2019 [13]. However, there is limited information about e-cigarette use among students aged 13–15 years in the Philippines. We estimated the weighted prevalence of current and ever e-cigarette use and awareness in the Philippines and examined covariates associated with its use and awareness among students aged 13–15 years using data from the 2019 Global Youth Tobacco Survey (GYTS).

## 2. Materials and Methods

### 2.1. Data Source

The target population includes students aged 13–15 years who participated in the 2019 GYTS in the Philippines. The GYTS is a two-stage cross-sectional, nationally representative, school-based survey that is a global standard for systematically monitoring tobacco use by students and tracking key indicators of tobacco control [14]. GYTS uses a standardized methodology for school selection, class representation, and data processes. The two-stage sample design allows for classes at selected schools to be chosen randomly and for all students of those selected classes to be eligible for participation. To break each stage down further, the first stage comprises both public and private schools. These schools are chosen based on the probability proportional to the number of students at each school. For stage two, equal probability sampling is performed. Classes from each school are randomly selected, and all students in the selected classrooms are eligible for the survey. GYTS is an anonymous survey, and the questionnaire is self-administered by students [14].

The total sample comprised 12,479 students from 96 schools and 295 classes. Of the 295 classes that were chosen, 290 classes participated, resulting in 10,602 students completing the survey; of these, 6670 were aged 13–15 years. Thus, the overall response rate was 83.5%.

### 2.2. Measures

The outcomes of this study were current use, ever use, and awareness of e-cigarettes. Current use of e-cigarettes was defined as ≥1 day of e-cigarette use in the last 30 days, based on the response to the question “During the past 30 days, on how many days did you use electronic cigarettes?” Response options included 0 days, 1 or 2 days, 3 to 5 days, 6 to 9 days, 10 to 19 days, 20 to 29 days, and all 30 days. Ever use of e-cigarettes was defined as answering yes to the question “Have you ever tried or experimented with electronic cigarettes or e-cigarettes or vape, even one or two puffs?” Response options included yes and no. Awareness of e-cigarettes was defined as answering yes to the question “Before today, had you ever heard of or seen electronic cigarettes or e-cigarettes or vape?” Response options included yes and no.

Independent variables included age (13 years, 14 years, and 15 years), gender (boy, girl), grade level (7th, 8th, 9th, and 10th), school type (public, private), island group (Luzon, Visayas, and Mindanao), money spent on self in pesos (usually don’t have any spending money, less than P10.00, P10.00–P100.00, P101.00–P500.00, and more than P501.00), current use of any other tobacco products (yes, no), secondhand smoke exposure at home (yes, no), and secondhand smoke exposure at school or in other public places (yes, no). Any other tobacco products include cigarettes, any form of smoked tobacco products other than cigarettes such as cigars, waterpipes, pipes, and shisha, and any form of smokeless tobacco products such as snuff, chewing tobacco, dip, and betel quid with tobacco. Secondhand smoke exposure was defined as ≥1 day of smoke exposure in the last 7 days at home, in any enclosed public place, and in any outdoor public place. These included exposure to smoke at home and in any enclosed public place other than at home, such as schools, shops, restaurants, shopping malls, inside any public transportation vehicles, and any outdoor public place such as playgrounds, sidewalks, entrances to buildings, parks, and beaches.

### 2.3. Statistical Analyses

Weighted prevalence and 95% confidence intervals (CIs) for current use, ever use, and awareness of e-cigarettes were estimated among students aged 13–15 years in the Philippines. Chi-square tests were conducted to compare the differences in unadjusted prevalence between groups. Multivariable logistic regression models, controlling for sociodemographic characteristics, current use of other tobacco products, and secondhand smoke exposure, were used to examine factors associated with current use, ever use, and awareness of e-cigarettes. Adjusted odds ratios (aOR) and 95% CIs were reported. The covariates assessed included gender, grade level, school type, money spent on self weekly, island group, current use of any other tobacco products, and secondhand smoke exposure at home, school, and other public places. All analyses were conducted using SAS-callable SUDAAN v11.0.1 (RTI International, Research Triangle Park, NC, USA) to account for the complex sample design and generate nationally representative estimates.

## 3. Results

### 3.1. Characteristics of Students Who Currently Use, Ever Used, and Were Aware of E-Cigarettes

Of the 6670 students aged 13–15 years, about 80% attended a public school, 51.4% were girls, and 48.6% were boys. Approximately two in three students who currently (68.9%) or ever used (65.9%) e-cigarettes were boys (Table 1). Additionally, among students who currently used e-cigarettes, 29.6% were in the 8th grade and 36.3% were in the 9th grade.

The proportion of students who used any other tobacco products was 39.9% among those who currently use e-cigarettes, 30.5% among those who ever used e-cigarettes, and 13.9% among those who were aware of e-cigarettes.

Geographically, more than half (58.6%) of the students who participated in the survey and more than 40% (42.4%) of students who currently used e-cigarettes were in Luzon. Most students (70.6%) were not exposed to secondhand smoke at home, but more than half (59.9%) were exposed to secondhand smoke at school or any other public places. In addition, 43.5% of students who currently used e-cigarettes spent P10.00–P100.00 (pesos) during an average week.

### 3.2. Prevalence of Current Use, Ever Use, and Awareness of E-Cigarettes

Table 2 displays the weighted prevalence of current use, ever use, and awareness of e-cigarettes. Among students aged 13–15 years in the Philippines, the prevalence of current e-cigarette use was 14.1% (95% CI = 12.4%, 15.8%), ever use of e-cigarettes was 24.6% (95% CI = 22.4%, 26.9%), and awareness was 75.5% (95% CI = 73.0%, 78.0%). Among boys, the prevalence of current e-cigarette use was 20.9% (95% CI = 18.4%, 23.4%); for girls, the prevalence of current e-cigarette use was 7.5% (95% CI = 6.2%, 8.9%).

Boys had a significantly higher prevalence of current and ever use of e-cigarettes than girls (*p* < 0.001) (Table 2). Prevalence of current e-cigarette use, ever use of e-cigarettes, and awareness of e-cigarettes was significantly lower among students aged 13 years than those older (*p* < 0.001). Students living in Visayas had a significantly lower prevalence of current and ever use than students in other island groups (*p* < 0.001). Those who were exposed to secondhand smoke at home had a higher prevalence of current e-cigarette use at 20.5% (*p* < 0.001), ever use of e-cigarettes at 31.8% (*p* < 0.001), and awareness of e-cigarettes at 81.4% (*p* < 0.001) when compared to those who were not exposed (11.5%, 21.7%, and 73.2%, respectively) (*p* < 0.001). Similarly, those who were exposed to secondhand smoke at school or in other public places had a higher prevalence of current e-cigarette use (17.7%, *p* < 0.001), ever use of e-cigarettes (29.8%, *p* < 0.001), and awareness of e-cigarettes (82.3%, *p* < 0.001) when compared to those who were not exposed (8.7%, 17.0%, and 65.3%, respectively).

### 3.3. Factors Associated with Current Use, Ever Use, and Awareness of E-Cigarettes

After adjusting for covariates, current use of any smoked or smokeless tobacco products was associated with increased odds of current use of e-cigarettes (aOR = 5.2, 95% CI = 3.9, 6.9) and increased odds of ever use of e-cigarettes (aOR = 4.3, 95% CI = 3.2, 5.7). Secondhand smoke exposure at home and secondhand smoke exposure at school or in other public places were positively associated with current use, ever use, and awareness of e-cigarettes (Table 3). Relative to girls, boys had a significantly higher adjusted odds ratio of current use (aOR = 3.0, 95% CI = 2.5, 3.7), ever use (aOR = 3.3, 95% CI = 2.7, 4.0), and e-cigarette awareness (aOR = 1.3, 95% CI = 1.1, 1.5). Students living in the Luzon island group had higher adjusted odds of current use (aOR = 1.7, 95% CI = 1.2, 2.4), ever use (aOR = 1.6, 95% CI = 1.3, 2.2), and awareness (aOR = 1.8, 95% CI = 1.3, 2.4) of e-cigarettes, relative to those from the Visayas island group. Students living in Mindanao had 60% increased odds of current use of e-cigarettes than those living in Visayas (aOR = 1.6, 95% CI = 1.1, 2.5) (Table 3). Being in Grade 10 was positively associated with ever use of e-cigarettes (aOR = 2.1, 95% CI = 1.3, 3.5) compared to being in Grade 7. Additionally, students in Grade 7 had significantly lower adjusted odds of e-cigarette awareness relative to those in higher grades. Compared to students who did not have any spending money, those who spent more than P501.00 (pesos) were more likely to ever use (aOR = 2.0, 95% CI = 1.3, 3.2) e-cigarettes. Students who spent P101.00–P500.00 (pesos) had higher adjusted odds of ever use (aOR = 1.6, 95% CI = 1.1, 2.3) and awareness (aOR = 1.9, 95% CI = 1.4, 2.6) of e-cigarettes compared to those who did not have any spending money.

## 4. Discussion

The present study analyzed the most recent GYTS data for the Philippines and fills a research gap in understanding students’ e-cigarette use and its associated factors in the country. The overall prevalence of current e-cigarette use was 14.1%, ever use was 24.6%, and awareness was 75.5%. We found that current use of any smoked and smokeless tobacco products and exposure to secondhand smoke at home, school, or in other public places were associated with higher odds of current and ever e-cigarette use. Additionally, being a boy and living in the island group of Luzon or Mindanao were associated with current e-cigarette use.

Our findings are consistent with another study that examined e-cigarette use in 68 countries and found that students who were boys, students who currently smoked cigarettes or other tobacco products, and students exposed to secondhand smoke were more likely to use e-cigarettes [13]. Additionally, a cross-sectional study in Thailand, another country in the region, assessed the prevalence of e-cigarettes among adolescents and factors associated with e-cigarette use [15]. Patanavanich et al. found that the prevalence of current e-cigarette use in Thailand among seventh graders was 3.7% and ever use was 7.2% in 2019 [15,16]. These estimates are lower than our results of 11.7% for current e-cigarette use and 19.1% for ever use among seventh graders in the Philippines. The difference in current e-cigarette use between Thailand and the Philippines may partially be explained by Thailand’s prohibition on the importation and sale of e-cigarettes, which was enacted in 2015 [15,16]. Based on evidence from Southeast Asia, Thailand has some of the strongest anti-tobacco policies [17,18]. However, illicit trade may undermine these efforts [17,18].

We found that students aged 13–15 years who currently use any other tobacco product (e.g., cigarettes, cigars, waterpipes, pipes, shisha, snuff, chewing tobacco, dip, betel quid with tobacco) and those exposed to secondhand smoke at home, school, and any other public places were more likely to currently use e-cigarettes. Additionally, our study found that more self-spending was positively associated with ever e-cigarette use among students. This finding may imply that a higher range of disposable income in the Philippines encourages the use of e-cigarettes; previous studies have found that having disposable income facilitates access to tobacco products [19,20,21].

The Philippines have implemented e-cigarette legislation, such as the Rules and Regulations on Electronic Nicotine Delivery Systems (ENDS) [22,23] and the Republic Act No. 10643 [24], which requires health warnings on tobacco products. However, despite its well-intentioned purpose, ENDS reduced the minimum age for accessing vaping products from 21 to 18 years of age [25]. In May 2021, the Philippines adopted the Non-Combustible Nicotine Delivery Systems Regulation Act, which regulates the manufacture, distribution, sale, use, packaging, promotion, and advertisement of electronic nicotine systems, non-nicotine delivery systems, heated tobacco products, and novel tobacco products [26]. This law prohibits the sale of vaporized nicotine and non-nicotine products to an individual under the age of 18 years [26]. Additionally, the law prohibits the use of vaporized nicotine at any indoor public establishment, except in a Designated Vaping Area (DVA) [26]. This Act aligns with the WHO MPOWER, a set of evidence-based strategies for reducing the demand for tobacco, specifically advocating for enforced bans on tobacco advertising, promotion, and sponsorship strategy [11]. The Republic Act 11900, known as the Vaporized Nicotine and Non-Nicotine Products Regulation Act, became law on 25 July 2022, following rigorous scrutiny in the Philippine Congress. This legislation seeks to regulate e-cigarettes and offers cessation techniques to people who use tobacco. However, the law lowered the minimum age for accessing vaping products from 21 to 18 [25]. To evaluate the long-term effectiveness of this policy approach, continued surveillance of tobacco use, including e-cigarette use, is needed to track prevalence changes in the Philippines [27]. There is evidence that policy interventions to reduce youth tobacco use should be considered, including the MPOWER framework [2,11].

This study is subject to several limitations. Because this is a cross-sectional study, we cannot examine the causal effects between the covariates and outcome variables (e.g., current use, ever use, and awareness of e-cigarettes). This study is only generalizable to students aged 13–15 years who attended formal schooling, so we cannot extrapolate the results to those who may have dropped out of school, those who are not enrolled in school, or students in different age groups in the Philippines. Additionally, students may underreport due to their perceptions of social norms, which may have led to self-reporting bias and/or social desirability bias. Furthermore, information bias can arise from incorrect interpretation of questions and inaccurate answer selection, leading to misclassification of e-cigarette use. While the GTYS survey was conducted in 2019, youth usage patterns do also change rapidly.

## 5. Conclusions

The findings of this study indicate that as of 2019, approximately one in four Philippine students aged 13–15 years had ever used e-cigarettes and one in seven currently used e-cigarettes, respectively. Understanding the demographic characteristics, smoking exposures, and current tobacco use associated with e-cigarette use among students may help inform tobacco control strategies for reducing e-cigarette use. Continued surveillance of e-cigarettes can help monitor the effectiveness of current demand reduction strategies and may help identify new ones. Future research may be needed to evaluate the progress and impact of the Philippines’ policies on e-cigarette manufacture, distribution, sale, use, packaging, promotion, and advertisement.

## Figures and Tables

**Table 1 ijerph-20-07193-t001:** Characteristics of students aged 13–15 years (n = 6670) by current and ever e-cigarette users and e-cigarette awareness in the Philippines, Global Youth Tobacco Survey, 2019.

	Total	Current E-Cigarette Users	Ever E-Cigarette Users	E-Cigarette Awareness
	(n = 6670)	(n = 817)	(n = 1452)	(n = 4841)
	n (%) ^a^	n (%)	n (%)	n (%)
Gender				
Girls	3700 (51.4)	252 (31.1)	494 (34.1)	2638 (54.5)
Boys	2958 (48.6)	559 (68.9)	953 (65.9)	2198 (45.5)
Age (years)				
13	2222 (33.7)	208 (25.5)	369 (25.4)	1489 (30.8)
14	2139 (33.1)	265 (32.4)	461 (31.7)	1564 (30.2)
15	2309 (33.2)	344 (42.1)	622 (42.8)	1788 (39.0)
Island group				
Visayas	2186 (19.8)	196 (24.0)	367 (25.3)	1484 (30.7)
Luzon	2271 (58.6)	346 (42.4)	612 (42.1)	1847 (38.2)
Mindanao	2213 (21.6)	275 (33.7)	473 (32.6)	1510 (31.2)
Grade level				
Grade 7	674 (11.2)	77 (9.4)	116 (8.0)	376 (7.8)
Grade 8	2517 (36.1)	241 (29.6)	427 (29.6)	1680 (34.9)
Grade 9	2060 (32.8)	296 (36.3)	502 (34.8)	1573 (32.6)
Grade 10	1380 (19.9)	201 (24.7)	398 (27.6)	1189 (24.7)
School type				
Public	5391 (80.4)	638 (78.6)	1134 (78.6)	3868 (80.2)
Private	1244 (19.6)	174 (21.4)	309 (21.4)	952 (19.8)
Money spent on self				
Usually don’t have any spending money	281 (4.1)	35 (4.3)	55 (3.8)	183 (3.8)
Less than P10.00	1201 (16.1)	106 (13.1)	181 (12.5)	632 (13.1)
P10.00–P100.00	3267 (47.4)	353 (43.5)	636 (44.0)	2407 (49.8)
P101.00–P500.00	1649 (27.8)	260 (32.1)	480 (33.2)	1397 (28.9)
More than P501.00	250 (4.6)	57 (7.0)	92 (6.4)	210 (4.3)
Current use of any other tobacco products				
No	5430 (86.6)	439 (60.1)	899 (69.5)	3956 (86.1)
Yes	766 (13.4)	277 (39.9)	379 (30.5)	577 (13.9)
Secondhand smoke exposure at home				
No	4698 (70.6)	456 (56.0)	890 (61.5)	3303 (68.3)
Yes	1956 (29.4)	359 (44.0)	558 (38.5)	1533 (31.7)
Secondhand smoke exposure at school or any other public places				
No	2651 (40.1)	194 (23.7)	389 (26.8)	1640 (33.9)
Yes	4019 (59.9)	623 (76.3)	1063 (73.2)	3201 (66.1)

^a^ Sample size and weighted percentage.

**Table 2 ijerph-20-07193-t002:** Weighted prevalence of current and ever e-cigarette use and e-cigarette awareness among students aged 13–15 years in the Philippines, Global Youth Tobacco Survey, 2019.

	Current E-Cigarette Use	Ever E-Cigarette Use	E-Cigarette Awareness
	% (95% CI)	% (95% CI)	% (95% CI)
	14.1 (12.4–15.8)	24.6 (22.4–26.9)	75.5 (73.0–78.0)
Gender			
Girls ^b^	7.5 (6.2–8.9)	14.8 (12.5–17.0)	74.6 (71.4–77.9)
Boys	20.9 (18.4–23.4) ^a^	35.2 (32.4–37.9) ^a^	76.5 (74.1–79.0) ^a^
Age (years)			
13 ^b^	10.6 (8.6–12.6)	19.5 (16.6–22.3)	70.2 (65.9–74.4)
14	14.5 (12.1–16.8) ^a^	24.5 (21.3–27.7) ^a^	76.6 (73.8–79.4) ^a^
15	17.2 (14.4–20.0) ^a^	30.1 (26.8–33.3) ^a^	79.8 (77.0–82.6) ^a^
Island group			
Visayas ^b^	9.8 (7.8–11.9)	17.8 (14.6–21.0)	67.2 (61.6–72.8)
Luzon	15.9 (13.3–18.6) ^a^	27.8 (24.5–31.2) ^a^	81.4 (78.8–84.0) ^a^
Mindanao	12.9 (9.8–16.1) ^a^	22.1 (17.2–27.1) ^a^	67.1 (59.4–74.8)
Grade level			
Grade 7 ^b^	11.7 (8.9–14.6)	19.1 (15.9–22.3)	57.2 (50.6–63.8)
Grade 8	11.0 (9.3–12.7)	19.7 (17.1–22.4)	71.3 (67.8–74.9) ^a^
Grade 9	17.0 (14.5–19.5) ^a^	27.5 (24.2–30.8) ^a^	79.6 (76.9–82.3) ^a^
Grade 10	16.3 (12.7–19.8) ^a^	31.6 (27.4–35.8) ^a^	86.9 (84.5–89.4) ^a^
School type			
Private ^b^	14.2 (10.7–17.6)	25.9 (20.7–31.0)	79.0 (73.6–84.4)
Public	14.0 (12.3–15.8)	24.3 (21.9–26.7)	74.7 (72.4–77.0)
Money spent on self			
Usually don’t have any spending money ^b^	12.9 (9.3–16.6)	20.9 (15.8–25.9)	68.5 (63.1–73.8)
Less than P10.00	9.7 (7.2–12.1) ^a^	17.1 (13.8–20.3) ^a^	54.2 (49.6–58.8) ^a^
P10.00–P100.00	12.6 (10.5–14.6) ^a^	22.0 (19.6–24.5) ^a^	76.1 (73.5–78.7) ^a^
P101.00–P500.00	17.6 (14.5–20.8) ^a^	31.6 (28.0–35.2) ^a^	86.2 (83.9–88.5) ^a^
More than P501.00	23.0 (19.3–26.6) ^a^	37.5 (31.8–43.3) ^a^	86.7 (81.8–91.6) ^a^
Current use of any other tobacco products			
No ^b^	9.2 (7.6–10.7)	18.5 (16.4–20.6)	75.4 (72.7–78.1)
Yes	41.2 (35.7–46.6) ^a^	56.1 (50.4–61.7) ^a^	79.3 (75.6–82.9) ^a^
Secondhand smoke exposure at home			
No ^b^	11.5 (9.9–13.1)	21.7 (19.4–24.0)	73.2 (70.4–75.9)
Yes	20.5 (17.8–23.2) ^a^	31.8 (28.8–34.9) ^a^	81.4 (78.5–84.3) ^a^
Secondhand smoke exposure at school or any other public places			
No ^b^	8.7 (7.0–10.5)	17.0 (14.6–19.4)	65.3 (61.8–68.7)
Yes	17.7 (15.5–19.8) ^a^	29.8 (27.1–32.5) ^a^	82.3 (79.8–84.8) ^a^

^a^ Prevalence was statistically significantly different from the reference group at the *p* < 0.05 level using a chi-square test; ^b^ Reference group; Abbreviations: CI—Confidence interval.

**Table 3 ijerph-20-07193-t003:** Factors associated with current e-cigarette use, ever e-cigarette use, and e-cigarette awareness among students aged 13–15 years in the Philippines, Global Youth Tobacco Survey, 2019.

	Current E-Cigarette Use	Ever E-Cigarette Use	E-Cigarette Awareness
	AOR (95% CI)	AOR (95% CI)	AOR (95% CI)
Gender			
Girls	^a^	^a^	^a^
Boys	**3.0 (2.5–3.7)**	**3.3 (2.7–4.0)**	**1.3 (1.1–1.5)**
Island group			
Visayas	^a^	^a^	^a^
Luzon	**1.7 (1.2–2.4)**	**1.6 (1.3–2.2)**	**1.8 (1.3–2.4)**
Mindanao	**1.6 (1.1–2.5)**	1.5 (1.0–2.3)	1.1 (0.7–1.6)
Grade level			
Grade 7	^a^	^a^	^a^
Grade 8	1.1 (0.7–1.6)	1.2 (0.8–1.7)	**1.7 (1.2–2.4)**
Grade 9	**1.8 (1.1–2.9)**	**1.7 (1.2–2.6)**	**2.4 (1.7–3.3)**
Grade 10	1.6 (0.9–2.9)	**2.1 (1.3–3.5)**	**3.5 (2.5–4.8)**
School type			
Private	^a^	^a^	^a^
Public	1.0 (0.7–1.5)	1.0 (0.7–1.4)	0.9 (0.6–1.2)
Money spent on self			
Usually don’t have any spending	^a^	^a^	^a^
money			
Less than P10.00	0.6 (0.4–0.9)	0.7 (0.5–1.0)	0.5 (0.4–0.7)
P10.00–P100.00	0.8 (0.5–1.3)	1.0 (0.7–1.4)	1.1 (0.8–1.5)
P101.00–P500.00	1.2 (0.8–1.8)	**1.6 (1.1–2.3)**	**1.9 (1.4–2.6)**
More than P501.00	1.6 (0.9–2.8)	**2.0 (1.3–3.2)**	**1.9 (1.0–3.4)**
Current use of any other tobacco products			
No	^a^	^a^	^a^
Yes	**5.2 (3.9–6.9)**	**4.3 (3.2–5.7)**	1.0 (0.8–1.3)
Secondhand smoke exposure at home			
No	^a^	^a^	^a^
Yes	**1.4 (1.2–1.7)**	**1.3 (1.1–1.5)**	**1.3 (1.1–1.5)**
Secondhand smoke exposure at school or any other public places			
No	^a^	^a^	^a^
Yes	**1.7 (1.4–2.0)**	**1.8 (1.5–2.0)**	**2.1 (1.7–2.5)**

^a^ Reference value; Abbreviations: AOR—Adjusted odds ratio estimated from logistic regression controlling for all the covariates included in the table; CI—Confidence interval. Bold font indicates statistically significant results.

## Data Availability

Publicly available datasets were analyzed in this study. This data can be found here: https://nccd.cdc.gov/GTSS/rdPage.aspx?rdReport=OSH_GTSS.ExploreByLocation&islCountry=rp&islDataSource=GYTS&islLocation=451&islTopic=T03&islWHORegion=WPR&islYear=20152015&rdRnd=93116 (accessed on 3 March 2023).

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
