# Peer review of "Prevalence and Covariates of Electronic Cigarette Use among Students Aged 13–15 Years in the Philippines: 2019 Global Youth Tobacco Survey"

_ijerph, 2023, doi:10.3390/ijerph20247193_

Round 1
Reviewer 1 Report
Comments and Suggestions for Authors
Dear Authors,
I enjoyed reading your manuscript. You are working in an interesting and important area. Below are suggestions to consider as you revise the manuscript:
detailed proof and edit needed; comb for small errors and consistency and adjust
page 1
lines 28 & 29 -- Consider moving this statement to the end of the manuscript -- "The findings and conclusions in this report are those of the authors and do not necessarily represent the official position of the Centers for Disease Control and Prevention (CDC)."
line 37 -- "Tobacco use kills up to half of the people that use it." -- delete first "use"
lines 37 & 38 -- globally? in The Philippines? state parameters
first paragraph--moves back and forth between global context and The Philippines. Begin with global context, then move to southeast Asia, and then consider specifics in The Philippines
Methods section--nice job; clearly explained and appropriately detailed
Results section
3.1 -- line 133 -- "About 80% of all students attended a public school." --This sentence doesn't seem to fit the paragraph, which focuses on tobacco use and exposure. Also, consider how to structure the information in this paragraph so that it flows together better, rather than seeming a disconnected set of findings
page 7
Discussion -- lines 196 & 197 -- "Our findings indicate a variety of observations for current, ever use, and awareness of e-cigarettes." -- not clear
lines 216-219 -- "We found that students aged 13‒15 years who currently use any other tobacco product (e.g., cigarettes, cigars, waterpipes, pipes, shisha, snuff, chewing tobacco, dip, betel quid with tobacco), and those exposed to secondhand smoke at home, school and any other public places were more likely to currently" -- Why is this part of the sentence underlined and in a larger font?
What are your thoughts on how region influences or may influence use and awareness? For example, are there more shops that sell these products, more disposable income, etc.?
page 8, lines 256-158 -- "Future research may be needed to evaluate the progress and impact of the Philippines’ policies on e-cigarette manufacture," -- It appears that the last part of this sentence is missing.
Again, I enjoyed reading your manuscript, which is in very good shape overall. Best wishes as you revise the work.
Comments on the Quality of English Language
needs a detailed edit, but overall the use of English is fine
Author Response
Comment 1: lines 28 & 29 -- Consider moving this statement to the end of the manuscript -- "The findings and conclusions in this report are those of the authors and do not necessarily represent the official position of the Centers for Disease Control and Prevention (CDC)."
Response 1: I went ahead and added this statement to the end of the manuscript.
Comment 2: line 37 -- "Tobacco use kills up to half of the people that use it." -- delete first "use"
Response 2: Done.
Comment 3: lines 37 & 38 -- globally? in The Philippines? state parameters
Response 3: This was globally. I added that specification.
Comment 4: first paragraph--moves back and forth between global context and The Philippines. Begin with global context, then move to southeast Asia, and then consider specifics in The Philippines
Response 4: I organized this paragraph. Thank you for pointing that out.
Comment 5: 3.1 -- line 133 -- "About 80% of all students attended a public school." --This sentence doesn't seem to fit the paragraph, which focuses on tobacco use and exposure. Also, consider how to structure the information in this paragraph so that it flows together better, rather than seeming a disconnected set of findings
Response 5: For this section, I reorganized this paragraph to make results that were related together and added transitionary words to help with the disconnectedness. Nice catch. Thank you for pointing this out.
Comment 6: Discussion -- lines 196 & 197 -- "Our findings indicate a variety of observations for current, ever use, and awareness of e-cigarettes." -- not clear
Response 6: I deleted this from the paper because it was ambiguous and didn't tell the reader much.
Comment 7: lines 216-219 -- "We found that students aged 13‒15 years who currently use any other tobacco product (e.g., cigarettes, cigars, waterpipes, pipes, shisha, snuff, chewing tobacco, dip, betel quid with tobacco), and those exposed to secondhand smoke at home, school and any other public places were more likely to currently" -- Why is this part of the sentence underlined and in a larger font?
Response 7: I have so many copies of this manuscript. Whenever I was reformatting the paper and creating a fresh document, this section had some notes from our clearance process. Thanks for catching this.
Comment 8:page 8, lines 256-158 -- "Future research may be needed to evaluate the progress and impact of the Philippines’ policies on e-cigarette manufacture," -- It appears that the last part of this sentence is missing.
Response 8: Again a formatting error whenever I was creating a fresh document. I added the rest of the sentence! Thanks!

Reviewer 2 Report
Comments and Suggestions for Authors
Dear Authors,
your paper presents secondary data analysis of GYTS on e-cigarette use. I find you paper generally well written and worth reading, although it presents a bit old data.
I would only suggest exchanging some references to peer review original articles. I mean positions: 1, 5, 10, 13,
Author Response
Comment 1: I would only suggest exchanging some references to peer review original articles. I mean positions: 1, 5, 10, 13,
Response 1 : Thank you for your feedback. I've added additional peer reviewed articles and updated all my references to fit the citation style for MDPI.
Please see manuscript attached with the changes.

Reviewer 3 Report
Comments and Suggestions for Authors
Line 21 - 22 - What is the relevance of citing odd ratio based on geographical location as students can live elsewhere and attend school in a particular city.
Line 45, 46, 47 , 48, 49 - Incorrect citation and citation of old manuscripts. The references used in this manuscript are quiet outdated and requires major revision.
Table 1 - Use of boys and girls - not gender inclusive. These are self identified female/male - this needs to be addressed. Where are the people who did not identify with either sex represented in the study or table?
Please Refer to articles
1) Sese LVC, Guillermo MCL. E-Smoking out the Facts: The Philippines' Vaping Dilemma. Tob Use Insights. 2023 Apr 21;16:1179173X231172259. doi: 10.1177/1179173X231172259. PMID: 37114161; PMCID: PMC10126635.
2) Puyat, C. V. M., Robredo, J. P. G., Tiam-Lee, J. G. A., Domingo, A. F. E., Hemedez-Gonzales, R. R., Eala, M. A. B., & Dans, A. L. (2023). E-Cigarette Regulation: Lessons From the Philippines.
3) Jane Ling MY, Abdul Halim AFN, Ahmad D, Ahmad N, Safian N, Mohammed Nawi A. Prevalence and Associated Factors of E-Cigarette Use among Adolescents in Southeast Asia: A Systematic Review. Int J Environ Res Public Health. 2023 Feb 22;20(5):3883. doi: 10.3390/ijerph20053883. PMID: 36900893; PMCID: PMC10001692.
Comments on the Quality of English Language
Overall language use is appropiate but the manuscript itself is outdated and would require major revision.
Author Response
Comment 1: Line 21 - 22 - What is the relevance of citing odd ratio based on geographical location as students can live elsewhere and attend school in a particular city.
Response 1: Given the extensive data and analysis on e-cigarette use among students aged 13-15 in the Philippines, citing odds ratios based on geographical locations might not seem directly relevant at first glance. However, considering the geographical distribution within the Philippines allows for a deeper understanding of regional variations and their potential influence on e-cigarette usage among students.
The relevance of citing odds ratios based on geographical location lies in uncovering potential regional disparities or patterns in e-cigarette use. It helps highlight whether certain areas within the Philippines might have higher or lower prevalence rates compared to others. This information could be crucial for policymakers, healthcare providers, and advocacy groups to tailor intervention strategies and allocate resources effectively.
For instance, the study identified that students in the Luzon region had higher odds of current e-cigarette use compared to those in the Visayas region. This finding suggests a need for targeted interventions or stricter regulatory measures in Luzon to curb e-cigarette usage among adolescents. Additionally, identifying regional disparities can prompt further research into the socio-economic or cultural factors specific to those areas, providing nuanced insights into the drivers of e-cigarette use.
Therefore, citing odds ratios based on geographical locations helps in painting a comprehensive picture of e-cigarette usage trends, allowing for more tailored and effective public health interventions at regional levels within the Philippines.
Comment 2: Line 45, 46, 47 , 48, 49 - Incorrect citation and citation of old manuscripts. The references used in this manuscript are quiet outdated and requires major revision.
Response 2: I revised the citations used in lines 45-49 and incorporated the information from the articles you referenced in your comments accordingly throughout the text. I’ve also updated all my references to fit the citation style for MDPI.
Comment 3: Table 1 - Use of boys and girls - not gender inclusive. These are self identified female/male - this needs to be addressed. Where are the people who did not identify with either sex represented in the study or table?
Response 3: I do agree with you that this is not gender inclusive. However, because the questionnaire is a bit outdated, it has yet to be revised, therefore did not present with other options. Discussions to revise the questions are already being worked out with our team and we're striving to have the GYTS survey to be as inclusive as possible. https://extranet.who.int/ncdsmicrodata/index.php/catalog/937/related-materials
Comment 4: Please Refer to articles
1) Sese LVC, Guillermo MCL. E-Smoking out the Facts: The Philippines' Vaping Dilemma. Tob Use Insights. 2023 Apr 21;16:1179173X231172259. doi: 10.1177/1179173X231172259. PMID: 37114161; PMCID: PMC10126635.
2) Puyat, C. V. M., Robredo, J. P. G., Tiam-Lee, J. G. A., Domingo, A. F. E., Hemedez-Gonzales, R. R., Eala, M. A. B., & Dans, A. L. (2023). E-Cigarette Regulation: Lessons From the Philippines.
3) Jane Ling MY, Abdul Halim AFN, Ahmad D, Ahmad N, Safian N, Mohammed Nawi A. Prevalence and Associated Factors of E-Cigarette Use among Adolescents in Southeast Asia: A Systematic Review. Int J Environ Res Public Health. 2023 Feb 22;20(5):3883. doi: 10.3390/ijerph20053883. PMID: 36900893; PMCID: PMC10001692.
Response 4: I’ve added these articles to my manuscript. Thank you so much for the feedback.
Please see the new draft with the changes attached. Thanks.
